Workshop at the 6th Symposium on Advances in Approximate Bayesian Inference (non-archival), 2024 1–12

# Hamiltonian Monte Carlo with categorical parameters using the Concrete distribution

**Jakob Torgander**                                    JAKOB.TORGANDER@STATISTIK.UU.SE
*Uppsala University*

**Måns Magnusson**                                    MANS.MAGNUSSON@STATISTIK.UU.SE
*Uppsala University*

**Jonas Wallin**                                    JONAS.WALLIN@STAT.LU.SE
*Lund University*

## Abstract

We introduce a method to enable Hamiltonian Monte Carlo (HMC) to simulate from mixed continuous and discrete posterior distributions. In particular, we show how the "Gumbel Max Trick" and the Concrete (Gumbel-softmax) distribution can be used for constructing a continuous approximation of a categorical distribution, and how this distribution can be efficiently implemented for HMC. We also illustrate how the Concrete distribution can be incorporated into a latent discrete parameter model, resulting in the Concrete Mixture model.

## 1. Introduction

Hamiltonian Monte Carlo (HMC) constitutes a powerful Markov Chain Monte Carlo (MCMC) technique for producing samples from probability distributions and is by many considered as the current state-of-art for Bayesian computation (Gelman et al., 2013; Betancourt, 2017). By utilising Hamiltonian dynamics in order to effectively explore the parameter space of a given posterior distribution, HMC provides an effective method for sampling from a wide range of complex and high-dimensional posteriors.

However, since Hamiltonian dynamics are defined by a system of differential equations, HMC requires the target posterior to be *continuous*. Consequently, mixed continuous and discrete parameter models are *not* directly accommodated by HMC. Historically, the methods for circumventing the problem of mixed continuous and discrete distribution for HMC can be categorised into the following two classes:

1. Separate sampling of the continuous and discrete parameters

2. Marginalisation of the discrete parameters.

The first class includes classical MCMC methods such as HMC-within-Gibbs (Neal, 2012), for which the continuous and discrete parameters are updated using separate Gibbs sampler steps. For another approach to this, (Zhou, 2020) provides a method in which the discrete parameters are sampled separately, but where the frequency of this sampling is controlled by an auxiliary parameter which is modelled and updated together with the continuous model parameters through standard HMC.

While the methods in this class *do* produce samples corresponding to both discrete and continuous parameters, they are subject to the well-known problem of producing highly correlated samples. This in turn might result in a less efficient exploration of the parameter space and hence in slow mixing times (Gelman et al., 2013).

In the second class of methods, the given mixed distribution is transformed into a fully continuous distribution through marginalisation of the discrete parameters, (Gelman et al., 2013). Conversely, the methods in the second class can here utilise the benefits of HMC to efficiently explore the parameter space, but do *not* directly produce discrete-valued samples. Additionally, the marginalisation procedure in general requires intractable manual computations before any model fitting and inference can be commenced.

In recent years, important work has been done on a *third* class of methods, where the discrete parameter space is turned into a continuous space using either relaxation or embedding techniques. As an example of this, (Nishimura et al., 2020) uses a continuous embedding to produce an *ordinal* discrete variable. Another important example of a continuous relaxation comes from the field of probabilistic machine learning with the Concrete (Gumbel-Softmax) distribution (Maddison et al., 2016; Jang et al., 2016). The Concrete distribution approximates a categorical distribution by utilising the so-called "Gumbel Max Trick". In the Gumbel Max Trick, the maximum of a sample of independent Gumbel(0,1)-distributed random variables is used to produce samples from an underlying categorical distribution.

The continuity of the Concrete distribution makes it a promising alternative to extend HMC to categorical distributions. A rigorous treatment of the distribution in the context of Bayesian computation and HMC has to our knowledge, however, not yet been done.

## 2. The Concrete Mixture (CM) model

The Concrete (Gumbel-Softmax) distribution (Maddison et al., 2016; Jang et al., 2016) has density function

$$p_Z(z_1, \ldots, z_K) = \Gamma(K) \tau^{K-1} \Big( \sum_{i=1}^{K} \pi_i / z_i^{\tau} \Big)^{-K} \prod_{i=1}^{K} (\pi_i / z_i^{\tau+1}), \tag{1}$$

where $\tau \in (0, \infty)$ is referred to as the *temperature* parameter of the distribution, and $\Gamma(\cdot)$ is the gamma function.

The Concrete distribution constitutes a continuous approximation of a categorical random variable $C$ with distribution $\pi = (\pi_1, \ldots, \pi_K)$, where $K$ is the number of categories (classes). As described in (Jang et al., 2016), when $\tau \to 0$, the vector $Z = (Z_1, \ldots, Z_K)$ approaches a one-hot vector, i.e. $Z_k \in \{0, 1\}$ and $Z_k = 1$ indicates the $k$-th class. The Concrete distribution is continuous with well-defined gradients for $z_i \in (0, 1)$ and is thus eligible for use within an HMC-framework.

A common situation in probabilistic modelling is where $C$ is a latent (unknown) parameter for a model, where data $x$ is assumed to have a conditional likelihood $f_k(x, \theta_k)$ given $C = k$. Here, each $f_k$ is in turn depending on a corresponding parameter vector $\theta_k$, on which we want to conduct inference. To model this situation using HMC, we now let $C$ be approximated by the Concrete-distributed variable $Z$ and then define the **Concrete**

**Mixture** (CM) model by the following (unnormalised) posterior distribution given a data point $x^{(i)}$,

$$p(\theta, z^{(i)}|x^{(i)}) \propto \Big( \sum_{k=1}^{K} z_k^{(i)} f_k(x^{(i)}, \theta_k) \Big) p(\theta) p_Z(z^{(i)}), \tag{2}$$

where $z^{(i)} = (z_1^{(i)}, \ldots, z_K^{(i)})$, $\theta = (\theta_1, \ldots, \theta_K)$, and $p(\theta)$ denotes some prior distribution for $\theta$. The CM model is constructed so that, for moderate values of $\tau$, the model resembles the classical mixture model on the corresponding form

$$p(\theta, \pi|x^{(i)}) \propto \Big( \sum_{k=1}^{K} \pi_k f_k(x^{(i)}, \theta_k) \Big) p(\theta) p(\pi), \tag{3}$$

where $\pi = (\pi_1, \ldots, \pi_K)$. However, as $\tau \to 0$, the CM model will approach the pure categorical model

$$p(\theta, c^{(i)}|x^{(i)}) \propto \Big( \sum_{k=1}^{K} \mathbb{1}(c^{(i)} = k) f_k(x^{(i)}, \theta_k) \Big) p(\theta) p(c^{(i)}), \tag{4}$$

where $c^{(i)} \in \{1, \ldots K\}$ indicates which class $x^{(i)}$ belongs to, and $p(c^{(i)})$ denotes the corresponding (categorical) prior distribution. Now, implementing the Concrete-distribution with HMC will, however, be subject to the problem of exploding gradients as the vector $z^{(i)}$ approaches the desired one-hot vector. This can easily be observed in Eq. (1) by the $K-1$ singularities formed when $z_i \to 0$ in the ratios $\pi_i/z_i$. This problem will in turn result in instability of the HMC-algorithm during the sampling process.

To mitigate the problem of exploding gradients, we propose that the Centred Gumbel (CG) distribution (Jang et al., 2016) instead be used when using HMC to sample from the Concrete distribution. The CG distribution has density function

$$p_U(u_1, \ldots, u_K) = \Gamma(K) \Big( \prod_{k=1}^{K} \exp(\pi_k - u_k) \Big) \Big( \sum_{k=1}^{K} \exp(\pi_k - u_k) \Big)^{-K}, \tag{5}$$

with the restriction $u_K = 0$. As shown by Jang et al. (2016), a CG-distributed random vector $U = (U_1, \ldots, U_K)$ can be transformed into a Concrete-distributed random vector $Z = (Z_1, \ldots, Z_K)$ through the transformation

$$Z_i = \frac{\exp(\frac{U_i}{\tau})}{1 + \sum_{j=1}^{K-1} \exp(\frac{U_j}{\tau})}. \tag{6}$$

It can further be shown (see Appendix B) that applying this transformation for the CM model results in the following corresponding unnormalised posterior

$$p(\theta, u^{(i)}|x^{(i)}) \propto \Big( \sum_{k=1}^{K} \frac{\exp(\frac{u_k^{(i)}}{\tau})}{1 + \sum_{j=1}^{K-1} \exp(\frac{u_j^{(i)}}{\tau})} f_k(x^{(i)}, \theta_k) \Big) p(\theta) p_U(u^{(i)}), \tag{7}$$

where $u^{(i)} = (u_1^{(i)}, \ldots, u_K^{(i)})$, and that this transformation circumvents the previously observed problem with exploding gradients. It can also be shown that this transformation,

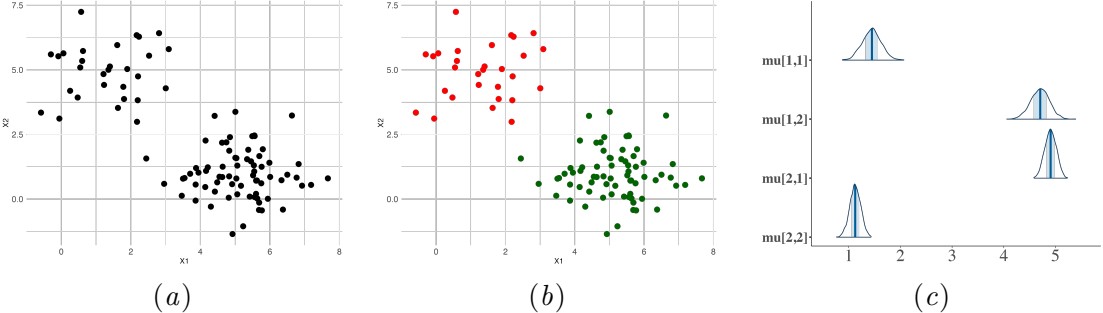

Figure 1: Sampling from the CM model. In (a) the original data from a bivariate Gaussian mixture is displayed. In (b) the class predictions formed by 10000 draws from the CM model are shown together with the posterior distribution for the mean parameters (c). The CM model clearly yields accurate samples corresponding to both categorical and continuous parameters.

however, has the side effect of introducing equivalent singularity problems when $\tau \to 0$ and that care thus must be taken when selecting this parameter. For more details of this, see Appendix B.

## 3. Experimental results

We will in this section provide a simple example on how the CM model introduced in the previous section can be used for modelling latent discrete parameters using HMC. For this purpose, we will use simulated data from a bivariate Gaussian mixture distribution, i.e. letting $f_k$ in Eq. (2) be the density of a 2-dimensional normal distribution with mean vector $\mu_k$, covariance matrix $\Sigma_k$ and class probabilities $\pi_k$. In particular, we will for this set of experiments use a data set consisting of 100 simulated data points with mean vectors $\mu_1 = (1, 5)$, $\mu_2 = (5, 1)$ and $\Sigma_1 = \Sigma_2 = I$, with $I$ denoting the $2 \times 2$ identity matrix. Moreover, we will here let the class distribution $\pi = (0.3, 0.7)$ and for simplicity enforce the resulting clusters to be linearly separable (see Figure 1(a)).

### 3.1. Parameter estimation

In the first experiment, we compare the ability of the CM model to accurately identify the true parameters of the previously defined mixture model. The experiments are conducted in R together with Stan (Carpenter et al., 2017), using the implementation in Eq. (7). To this model we let $\mu_i \sim N(2.5, 10)$ (i.e. centring the prior in between the two clusters in Figure 1(a)), and $\pi \sim Dirichlet(1, 1)$. Again for simplicity, we will here use a naive assumption of known unit covariance, i.e let $\Sigma = I$ be fixed for all $f_k$. We also let $\tau = 0.1$ be fixed for this experiment.

In Figure 1(b), we visualise the class predictions resulting from the simulated categorical variables $z^{(i)}$, and Figure 1(c) displays the posterior distribution corresponding to $\mu_i$. The figures are based on 10000 posterior draws. The class predictions $\hat{c}^{(i)}$ are formed for the

| (a) | | |
|---|---|---|
| **Method** | **1-Wasserstein** | **Runtime (s)** |
| HMC/CM | 7.0931 | 102.1689 |
| HMC/Margin. | 5.9171 | 18.9618 |
| Gibbs | 7.40021 | 2.3193 |

| (b) | | | |
|---|---|---|---|
| $\tau$ | $\mathbf{MSE}_{\hat{y}}$ | **Leapfrog steps** | **Step size** |
| 1 | 6.2651 | 15 | 0.3156 |
| 0.1 | 2.6842 | 31 | 0.1123 |
| 0.01 | 2.1293 | 267 | 0.0114 |
| 0.001 | 2.0707 | 1023 | 0.0015 |

Table 1: Comparison of actual and simulated parameters. In (a) the empirical 1-Wasserstein distances between the simulated and "true" reference posterior are compared between the CM model and the current recommended methods for sampling discrete parameters. In (b) the effect of the temperature parameter $\tau$ on the performance of the CM model is illustrated.

$i$-th data point through a majority vote, i.e. letting $\hat{c}^{(i)} = k$ if $z_k^{(i)}$ has the highest value when averaged over all draws.

As illustrated by these figures, the CM model manages to correctly classify the data points and results in posterior distributions correctly centred around their respective actual values.

Secondly, we compare the similarity of our generated samples to a "true" reference posterior generated by HMC, using the marginalised model in Eq. (3). This is done by computing the empirical 1-Wasserstein distance (Villani et al., 2009) between the samples from the CM model and samples from the marginalised model. We also include the corresponding distances using samples from standard Gibbs sampling, and from a *second* set of samples generated by marginalised HMC. The sampling was for these two methods carried out using Stan and JAGS (Plummer, 2004) respectively with the same prior distributions as for the CM model, using the R `transport` package (Schuhmacher et al., 2024) for computation of the Wasserstein distances.

The results of these experiments are collected in Table 1(a) and show that the CM model performs equivalently to the current standards of simulating when 1-Wasserstein distances are used as the similarity measure. Taking computational efficiency into account, measured by the sampler run-times in Table 1(a), we observe that the CM model however is significantly less efficient than both the HMC/marginalisation combination and the Gibbs sampler.

### 3.2. Temperature calibration

As previously mentioned in Section 2, the temperature parameter $\tau$ in Eq. (1) controls how closely the Concrete random variable approximates a categorical one hot vector. At the same time, as was observed in Eq. (7), when $\tau \to 0$ the CM model will suffer from similar singularity problems that was previously observed for the Concrete distribution.

To investigate the effect of the temperature on the CM model, we repeat the experiments from the previous section, now for varying values of $\tau$. In Table 1(b), we compare the MSE between actual (in-sample) data points and fitted values

$$\hat{y}^{(i)} = z_1^{(i)}\mu_1 + z_2^{(i)}\mu_2, \tag{8}$$

where $z_j^{(i)}$, $j = 1, 2$, denotes the simulated categorical variables from the CM model corresponding to the $i$-th data point. The intuition behind this comparison is that, the lower the temperature, the closer the CM model will be to the true (known) data generating process on average. To see the effect on the computational complexity, we include in this table the average number of leapfrog steps and the average step size of the underlying NUTS-sampler (Hoffman et al., 2014). We can from Table $1(b)$ observe a clear trade-off between predictive performance and computational cost as the temperature becomes lower.

## 4. Discussion

We have in this paper demonstrated the use of the Concrete distribution for simulating from mixed discrete and continuous parameter models. Although further research is needed for determining the scalability and reliability of the CM model, our experiments indicate that the model can be seen as an alternative for extending HMC to latent discrete parameter models.

As demonstrated in our experiments, the most significant bottleneck for the generalisation of the model is the computational complexity resulting from the difficult posterior geometries created by low temperatures, and by the fact that $K$ parameters are needed for each observation in the data set in order to model the individual $K$-dimensional approximated one-hot encoded vectors. This in turn will inevitably affect the scalability of the model as the number of data points or categories becomes large.

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

## Appendix A. The Concrete (Gumbel-Softmax) distribution

This section provides a brief introduction to the Concrete (Gumbel-Softmax) distribution and is based on the results and derivations in (Maddison et al., 2016; Jang et al., 2016).

Let $C$ denote a categorical variable with probability distribution $\pi = (\pi_1, \ldots, \pi_K)$ for some positive integer $K$. We here assume that the categories (classes) of $C$ are encoded with the integers $1, \ldots, K$. Furthermore, let $G_1, \ldots, G_K$ denote $K$ i.i.d. Gumbel$(0, 1)$-variables, each having density function

$$p_{G_i}(x) = e^{-e^{-x}}. \tag{9}$$

The Gumbel Max trick then uses samples from the variables $G_i$ in order to sample from $C$ using the following procedure:

1. Generate samples $g_1, \ldots, g_K$ from $G_1, \ldots, G_K$

2. Let $x = \underset{i}{\operatorname{argmax}}(\log \pi_i + g_i)$

Letting $X$ denote the random variable resulting from this procedure, it can be shown Gumbel (1954) that $\mathbb{P}(X = k) = \pi_k$, i.e. the distribution of $X$ coincides with the distribution of $C$. Now, as the argmax-function can be shown to be problematic to optimise, the idea in (Maddison et al., 2016) is to approximate the argmax function using the *softmax* function

$$\operatorname{softmax}(x)_i = \frac{\exp(x_i)}{\sum_j \exp(x_j)}$$

using the following transformation.

$$Z_i = \frac{\exp(\log \pi_i + G_i)/\tau}{\sum_{j=1} \exp(\log \pi_j + G_j)/\tau}. \tag{10}$$

Here, $\tau \in (0, \infty)$ is referred to as the *temperature* parameter of the distribution. Now, as $\tau \to 0$, the vector $Y = (Z_1, \ldots, Z_K)$ approaches a one-hot vector, where $Z_k = 1$ is equivalent to $C = k$. Conducting the transformation in Eq. (10) then yields the **Concrete** or **Gumbel-Softmax** distribution with density function

$$p_Z(z_1, \ldots, z_K) = \Gamma(K)\tau^{K-1}\Big(\sum_{i=1}^{K} \pi_i/z_i^{\tau}\Big)^{-K} \prod_{i=1}^{K}(\pi_i/z_i^{\tau+1}), \tag{11}$$

with parameters $\pi, \tau, z_i \in (0, 1)$ and with $\sum z_i = 1$.

## Appendix B. Theoretical results

### B.1. The exploding gradients problem of the CM model

In this section we will show that the CM model Eq. (2) suffers from the problem of exploding gradients but that these problems are removed if the transformed model in Eq. (7) instead is used. We also show that the transformed model however introduces a new problem with exploding gradients in the parameter $\tau$.

To this end, we first observe that the log-posterior of the CM model in Eq. (2) can for a single observation $x$ and up to the additive constants $C_1, C_2$, be written as

$$l_{\mathrm{CM}}(z_1, \ldots, z_K) = \log\Big(\sum_{k=1}^{K} z_k f_k(x, \theta_k)\Big) + \log(p_Z(z)) + C_1, \tag{12}$$

$$= \log\Big(\sum_{k=1}^{K} z_k f_k(x, \theta_k)\Big) + \log\Big(\Gamma(K)\tau^{K-1}\Big(\sum_{k=1}^{K} \pi_k/z_k^\tau\Big)^{-K} \prod_{i=1}^{K}\Big(\frac{\pi_k}{z_k^{\tau+1}}\Big)\Big) + C_1 \tag{13}$$

$$= \log\Big(\sum_{k=1}^{K} z_k f_k(x, \theta_k)\Big) - K\log\Big(\sum_{k=1}^{K} \frac{\pi_k}{z_k^\tau}\Big) + \sum_{k=1}^{K}\log\Big(\frac{\pi_k}{z_k^{\tau+1}}\Big) + C_2. \tag{14}$$

To see that the CM model suffers from exploding gradients, we observe that the partial derivatives of $l_{\mathrm{CM}}$ with respect to $z, \pi$ can be written as follows,

$$\frac{\partial l_{\mathrm{CM}}}{\partial z_j} = \frac{f_j(x, \theta_j)}{\sum_{k=1}^{K} z_k f_k(x, \theta_k)} - \frac{K}{\sum_{k=1}^{K} \frac{\pi_k}{z_i^\tau}} \cdot \Big(\frac{-\pi_j \tau}{z_j^{\tau+1}}\Big) + \frac{1}{\frac{\pi_j}{z_j^{\tau+1}}} \cdot \Big(-\frac{(\tau+1)\pi_j}{z_j^{\tau+2}}\Big)$$

$$= \frac{f_j(x, \theta_j)}{\sum_{k=1}^{K} z_k f_k(x, \theta_k)} + \frac{K\pi_j \tau}{\sum_{k=1}^{K} \frac{\pi_k}{z_i^\tau}} \cdot \Big(\frac{1}{z_j^{\tau+1}}\Big) - \frac{z_j^{\tau+1}}{\pi_j} \cdot \frac{(\tau+1)\pi_j}{z_j^{\tau+2}} \tag{15}$$

$$= \frac{f_j(x, \theta_j)}{\sum_{k=1}^{K} z_k f_k(x, \theta_k)} + \frac{K\pi_j \tau}{\sum_{k=1}^{K} \frac{\pi_i}{z_i^\tau}} \cdot \frac{1}{z_j^{\tau+1}} - \frac{\tau+1}{z_j},$$

and,

$$\frac{\partial l_{\mathrm{CM}}}{\partial \pi_k} = \frac{-K}{\sum_{k=1}^{K} \frac{\pi_k}{z_j^\tau}} \cdot \Big(\frac{1}{z_j^\tau}\Big) + \frac{z_j^{\tau+1}}{\pi_j} \cdot \Big(\frac{1}{z_k^{\tau+1}}\Big)$$

$$= \frac{1}{\pi_j} - \frac{K}{\sum_{k=1}^{K} \frac{\pi_k}{z_k^\tau}} \cdot \frac{1}{z_j^\tau} \tag{16}$$

Observing that $\frac{1}{z_j} \to \infty$ as $z_j \to 0$, we can conclude from the expressions (15)-(16) that the CM model indeed will suffer from exploding gradients when $z$ approaches a one-hot vector.

Conversely, the log-posterior of the transformed CM model in Eq. (7) can similarly be written,

$$l_{\text{CGM}}(u_1, \ldots, u_K) = \log \Big( \sum_{k=1}^{K} \frac{\exp(\frac{u_k}{\tau})}{1 + \sum_{i=1}^{K-1} \exp(\frac{u_i}{\tau})} f_k(x, \theta_k) \Big) + \log(p_U(u)) + C_3$$

$$= \log \Big( \sum_{k=1}^{K} \frac{\exp(\frac{u_k}{\tau})}{1 + \sum_{i=1}^{K-1} \exp(\frac{u_i}{\tau})} f_k(x, \theta_k) \Big)$$

$$+ \sum_{k=1}^{K} \log \Big( \exp(\pi_k - u_k) \Big) - K \log \Big( \sum_{k=1}^{K} \exp(\pi_k - u_k) \Big) + C_4$$

$$= \log \Big( \sum_{k=1}^{K} \frac{\exp(\frac{u_k}{\tau})}{1 + \sum_{i=1}^{K-1} \exp(\frac{u_i}{\tau})} f_k(x, \theta_k) \Big) + \sum_{k=1}^{K} (\pi_k - u_k) - K \log \Big( \sum_{k=1}^{K} \exp(\pi_k - u_k) \Big) + C_4.$$

$$(17)$$

The corresponding partial derivatives with respect to $\pi$ are here given by

$$\frac{\partial l_{\text{CGM}}}{\partial \pi_j} = \pi_j - K \frac{\exp(\pi_j - u_j)}{\sum_{k=1}^{K} \exp(\pi_k - u_k)}. \tag{18}$$

Since, for any $i$, $\frac{\exp(\pi_j - u_j)}{\sum_{k=1}^{K} \exp(\pi_i - u_i)} \leq 1$, it is clear that our transformation alleviates the exploding gradients problem in the $\pi$-directions. Now, for the partial derivatives with respect to $u$, we first focus on the first term of Eq. (17) and denote this by $s(u)$. Furthermore, we use the fact that $u_K = 0$ to receive the following simplifying notation

$$1 + \sum_{k=1}^{K-1} \exp(\frac{u_k}{\tau}) = \sum_{k=1}^{K} \exp(\frac{u_k}{\tau}). \tag{19}$$

The partial derivatives of $s(u)$ can then be computed as follows:

$$\frac{\partial s}{\partial u_j} = \frac{1}{\sum_{k=1}^{K} \frac{\exp(\frac{u_k}{\tau})}{1 + \sum_{i=1}^{K-1} \exp(\frac{u_i}{\tau})} f_k(x, \theta_k)} \cdot \Bigg( \frac{\frac{1}{\tau} \exp(\frac{u_j}{\tau}) \Big( \sum_{k=1}^{K} \exp(\frac{u_k}{\tau}) \Big) - \exp(\frac{u_j}{\tau}) \frac{1}{\tau} \exp(\frac{u_j}{\tau})}{(\sum_{k=1}^{K} \exp(\frac{u_k}{\tau}))^2} f_j(x, \theta_j)$$

$$- \sum_{i \neq j} \frac{\exp(\frac{u_i}{\tau})}{\Big( \sum_{k=1}^{K} \exp(\frac{u_k}{\tau}) \Big)^2} \frac{1}{\tau} \exp(\frac{u_j}{\tau}) f_i(x, \theta_i) \Bigg)$$

$$= \frac{1}{\sum_{k=1}^{K} \frac{\exp(\frac{u_k}{\tau})}{\sum_{k=1}^{K} \exp(\frac{u_k}{\tau})} f_k(x, \theta_k)} \cdot \Bigg( \frac{\frac{1}{\tau} \exp(\frac{u_j}{\tau})}{\sum_{k=1}^{K} \exp(\frac{u_k}{\tau})} \Big( (1 - \frac{\exp(\frac{u_j}{\tau})}{\sum_{k=1}^{K} \exp(\frac{u_k}{\tau})}) f_j(x, \theta_j)$$

$$- \sum_{i \neq j} \frac{\exp(\frac{u_i}{\tau})}{\sum_{k=1}^{K} \exp(\frac{u_k}{\tau})} f_i(x, \theta_i) \Big) \Bigg)$$

$$
\begin{aligned}
&= \frac{\frac{1}{\tau}\exp(\frac{u_j}{\tau})}{\sum_{k=1}^{K}\exp(\frac{u_k}{\tau})f_k(x,\theta_k)} \cdot \left( \frac{\sum_{i\neq j}\exp(\frac{u_i}{\tau})}{\sum_{k=1}^{K}\exp(\frac{u_k}{\tau})}f_j(x,\theta_j) - \frac{\sum_{i\neq j}\exp(\frac{u_i}{\tau})f_i(x,\theta_i)}{\sum_{k=1}^{K}\exp(\frac{u_k}{\tau})} \right) \\
&= \frac{\frac{1}{\tau}\exp(\frac{u_j}{\tau})}{\sum_{k=1}^{K}\exp(\frac{u_k}{\tau})} \cdot \left( \frac{\sum_{i\neq j}\exp(\frac{u_i}{\tau})(f_j(x,\theta_j)-f_i(x,\theta_i))}{\sum_{k=1}^{K}\exp(\frac{u_k}{\tau})f_k(x,\theta_k)} \right) \\
&= \frac{\frac{1}{\tau}\exp(\frac{u_j}{\tau})}{\sum_{k=1}^{K}\exp(\frac{u_k}{\tau})} \cdot \left( \frac{\sum_{i=1}^{K}\exp(\frac{u_i}{\tau})(f_j(x,\theta_j)-f_i(x,\theta_i))}{\sum_{k=1}^{K}\exp(\frac{u_k}{\tau})f_k(x,\theta_k)} \right),
\end{aligned}
\tag{20}
$$

where the last equality follows since $\exp(\frac{u_i}{\tau})(f_j(x,\theta_j)-f_i(x,\theta_i))=0$ for $i=j$. Since the corresponding partial derivatives for the remaining terms of Eq. (17) now can be written

$$
-1-K\frac{-\exp(\pi_j-u_j)}{\sum_{k=1}^{K}\exp(\pi_k-u_k)} = -1+K\frac{\exp(\pi_j-u_j)}{\sum_{k=1}^{K}\exp(\pi_k-u_k)},
\tag{21}
$$

we thus receive the following expression for $\frac{\partial l_{\mathrm{CGM}}}{\partial u_j}$

$$
\frac{\frac{1}{\tau}\exp(\frac{u_j}{\tau})}{\sum_{k=1}^{K}\exp(\frac{u_k}{\tau})} \cdot \left( \frac{\sum_{i=1}^{K}\exp(\frac{u_i}{\tau})(f_j(x,\theta_j)-f_i(x,\theta_i))}{\sum_{k=1}^{K}\exp(\frac{u_k}{\tau})f_k(x,\theta_k)} \right) - 1 + K\frac{\exp(\pi_j-u_j)}{\sum_{k=1}^{K}\exp(\pi_k-u_k)}.
\tag{22}
$$

Lastly, since the following upper bound can easily be seen

$$
\left| \frac{\partial l_{\mathrm{CGM}}}{\partial u_j} \right| \leq \left| \frac{1}{\tau} \right| \cdot \left| \frac{\sum_{i=1}^{K}\exp(\frac{u_k}{\tau})(f_j(x,\theta_j)-f_i(x,\theta_i))}{\sum_{k=1}^{K}\exp(\frac{u_k}{\tau})f_k(x,\theta_k)} \right| + (K+1)
\tag{23}
$$

$$
\leq \frac{1}{\tau} \cdot \left( \frac{\sum_{i=1}^{K}\exp(\frac{u_i}{\tau})f_j(x,\theta_j)}{\sum_{k=1}^{K}\exp(\frac{u_k}{\tau})f_k(x,\theta_k)} + 1 \right) + (K+1)
\tag{24}
$$

$$
\leq \frac{1}{\tau} \cdot \left( \frac{f_j(x,\theta_j)}{\min_{0\leq k\leq K} f_k(x,\theta_k)} + 1 \right) + (K+1),
\tag{25}
$$

we can conclude that the transformation to the CM model also solves the problem of exploding gradients in the $u_j$-directions. We can from Eq. (17) however observe that $\frac{\partial l_{\mathrm{CGM}}}{\partial u_j}$ becomes unbounded as $\tau \to 0$ for which $\exp(\frac{u_j}{\tau}) \to \infty$ for all $i$. Thus, the transformation to the CM model introduces an equivalent problem of exploding gradients in the $\tau$-direction. Using the transformed CM model with HMC, we hence receive a trade-off between computational stability and the distance from the Concrete approximation to the true underlying categorical distribution.

## B.2. Proof of Eq. (10) (transformation of the CM model)

We will in this section prove that the transformation in Eq. (10) indeed transforms the CM model to the unconstrained model in Eq. (7). For this purpose, let $Z=(Z_1,\dots,Z_K)$ be Concrete distributed with parameter $\tau$ fixed. We then define the transformation $H(Z)$ component-wise as follows

$$
U_i = H(Z)_i = \tau(\log Z_i - \log Z_K), \quad i=1,\dots,K.
\tag{26}
$$

It can be easily confirmed that this transformation is one-to-one and that

$$Z_i = h^{-1}(U) = \frac{\exp(\frac{U_i}{\tau})}{1 + \sum_{i=1}^{K-1} \exp(\frac{U_i}{\tau})} \tag{27}$$

Now, by the change of variables formula,

$$\begin{aligned}\Big(\sum_{k=1}^{K} z_k f_k(x, \theta_k)\Big) p(\theta) p_Z(z) &= \Big(\sum_{k=1}^{K} h^{-1}(u) f_k(x, \theta_k)\Big) p(\theta) p_Z(h^{-1}(u)) \cdot |J| \\ &= \Big(\sum_{k=1}^{K} \frac{\exp(\frac{u_k}{\tau})}{1 + \sum_{j=1}^{K-1} \exp(\frac{u_j}{\tau})} f_k(x, \theta_k)\Big) p(\theta) p_Z(h^{-1}(u)) \cdot |J|,\end{aligned} \tag{28}$$

where $|J|$ denotes the determinant of the Jacobian $J$ corresponding to the transformation. Now, when constructing the Concrete distribution, Jang et al. (2016) show that letting $U = (U_1, \ldots, U_K)$ follow the Centred Gumbel distribution defined in Eq. (5), the following result holds

$$p_Z(z) = p_U(h(z)) \cdot |J'|,$$

where $J'$ again denotes the corresponding Jacobian. Since the transformation $h$ is one to one, the inverse of $J'$ exists and hence

$$p_U(u) = \frac{1}{|J'|} \cdot p_Z(h^{-1}(u)) = |J'^{-1}| \cdot p_Z(h^{-1}(u)). \tag{29}$$

Comparing this identity with Eq. (28) we see that $J = J'^{-1}$ and in turn

$$\Big(\sum_{k=1}^{K} z_k f_k(x, \theta_k)\Big) p(\theta) p_Z(z) = \Big(\sum_{i=1}^{K} \frac{\exp(\frac{u_i}{\tau})}{1 + \sum_{j=1}^{K-1} \exp(\frac{u_j}{\tau})} \cdot f_k(x, \theta_k)\Big) p(\theta) p_U(u). \tag{30}$$

This completes the proof.

