# OpenReview forum: "Hamiltonian Monte Carlo with categorical parameters using the Concrete distribution"
_approximateinference.org/AABI/2024/Symposium — AABI 2024_

### Official Review · Reviewer_6kUv · 2024-04-07
**This paper tests the ”Gumbel Max Trick” in the context of HMC**

**Rating:** 5
**Confidence:** 3

**Review:**

This paper incorporates the "Gumbel Max Trick" into the framework of Hamiltonian Monte Carlo (HMC) to simulate from mixed continuous and discrete posterior distributions. The authors evaluate the performance of the resulting algorithm through a toy experiment, revealing that it is not as computationally efficient as standard existing methods.

I have two major concerns as follows:
1. The ”Gumbel Max Trick” has been widely used for constructing a continuous approximation of a categorical discrete distribution, especially in the deep learning community. See, e.g., the following papers published in ICLR 2017.

@article{maddison2016concrete,

  title={The concrete distribution: A continuous relaxation of discrete random variables},

  author={Maddison, Chris J and Mnih, Andriy and Teh, Yee Whye},

  journal={arXiv preprint arXiv:1611.00712},

  year={2016}

}

@article{jang2016categorical,

  title={Categorical reparameterization with gumbel-softmax},

  author={Jang, Eric and Gu, Shixiang and Poole, Ben},

  journal={arXiv preprint arXiv:1611.01144},

  year={2016}

}

Therefore, the main concept presented in this work is not sufficiently novel.

2. The experimental section is weak, as it only includes a very simple toy example, and lacks comparison with other related HMC methods, such as the following one.

@article{zhang2012continuous,

  title={Continuous relaxations for discrete hamiltonian monte carlo},

  author={Zhang, Yichuan and Ghahramani, Zoubin and Storkey, Amos J and Sutton, Charles},

  journal={Advances in Neural Information Processing Systems},

  volume={25},

  year={2012}

}

---

### Official Review · Reviewer_SZgz · 2024-04-22
**Natural idea of handling discrete random variables in HMC, with a good analysis on the size of gradients**

**Rating:** 7
**Confidence:** 4

**Review:**

In this paper, the authors propose to handle discrete random variables in HMC by relaxing those random variables to continuous random variables with the concrete distribution. They show that a straightforward approach for implementing this idea has an issue of exploding gradient, and suggest to use an alternative parameterisation of the concrete distribution, which does not suffer from the same exploding-gradient issue although it still has a similar problem with respect to the so called temperature parameter of the concrete distribution. The experiments show that in terms of accuracy, the authors' approach is comparable with existing approaches based on Gibbs sampling and marginalisation, although it is much slower than these competitors.

The experimental results in the paper are slightly negative, but I still support the acceptance of the paper. The main idea of the paper, which is to relax discrete random variables to continuous ones with the concrete distribution, is something that would occur to the practitioners of HMC very naturally. But as the paper claims, this natural and somewhat obvious idea hasn't been tried successfully before. The authors gives one plausible explanation on why this is the case; they describe the exploding-gradient issue of a naive implementation of the idea. The authors then propose a solution for this exploding-gradient problem that relies on the reparameterisation of the concrete distribution. I found both the identification of this exploding-gradient problem and the authors' solution very interesting. I think that the identification and the solution both will be useful not just for extending HMC to cover discrete random variables, but also for analysing the use of the concrete distribution in different applications.

Here are some minor comments.

* page 2: I would change the subscripts of z so that it is indexed by both i and k. I think that the model assigns one z for each observed data point x_i.
* equation (13), page 8: pi_k \ z_k^\tau ===> pi_k / z_k^\tau
* right above equation (19), page 9: u_1 = 0 ===> u_K = 0
* equation (20), page 10: I think that the exp(u_i)/tau factor is missing in sum f_j(..) - f_i(..).
* equation (21), page 10: I think that u_j should be 1.
* equation (22), page 10: I think that the formula here has the two problems mentioned above.
* equation (23): Please double-check the upper bound.

---

### Official Review · Reviewer_CU4F · 2024-04-23

**Rating:** 7
**Confidence:** 4

**Review:**

This paper presents a framework for HM sampling with mixed continuous-categorical distributions. The key idea is to use the Centred Gumbel (CG) distribution to sample from the Concrete Gumbel-Softmax distribution -- where the Concrete distribution is used to approximate the categorical distribution. The paper presents initial results of the suitability of this approach for continuous-categorical sampling by evaluating: parameter estimation quality, and the effect of the temperature parameter.
While the framework is parsimonious, the contribution could be useful and the implications wide. Thus, I believe the idea has potential of a greater impact.

---

### Official Review · Reviewer_4btU · 2024-04-24
**Using Concrete Distribution to sample discrete parameters with HMC**

**Rating:** 6
**Confidence:** 2

**Review:**

This work investigates the usage of the Concrete distribution as a means of sampling discrete parameters using Hamiltonian Monte Carlo (HMC).

In doing so, the authors also propose an alternative parameterization of the Gubeml distribution, which they refer to as the *Centered Gumbel*, as a way of mitigating the exploding gradient issues that arise when working with the Gumbel as the relaxed variables $(z_i) \to 0$. But, as the authors point out, this only shifts the problem of exploding gradients into the temperature $\tau$ as $\tau \to 0$, which is of course related to the variables $z_i \approx 0$. But it indeed seems like a good idea in the context of HMC, and so overall is an interesting idea.

In the empirical investigation, the authors focus a lot on the Mean-Squared-Error (MSE) from the true parameters; it is somewhat unclear to me that this is indeed a good metric to focus on in the contet of *sampling*, where we are interested in approximating *expectations* wrt. the distribution.

But overall, I think the usage of the Concrete distribution in the context of HMC sampling is an interesting idea, and I believe this work to be suitable for this venue.

---

### Official Review · Reviewer_xWfA · 2024-04-24
**solid work**

**Rating:** 7
**Confidence:** 4

**Review:**

The paper propose a straight forward way of relaxation categorical r.v. in order to use HMC. While idea is straightforward, all the details and implementation is a huge amount of work. So I appreciate this paper come, hope it will be used by practitioners. Also only good words can be said about writing and presentation of paper in general.

---

### Official Review · Reviewer_XXJ2 · 2024-04-24
**Interesting tackling of Hamiltonian Monte Carlo but flawed performance metrics**

**Rating:** 7
**Confidence:** 4

**Review:**

The authors tackle the problem of using Hamiltonian Monte Carlo for posterior sampling when the posterior includes discrete parameters. This is a well-known problem with several proposed solutions, but any new idea is welcome and worth investigating.

The solution presented here relies on a continuous approximation of the discrete categorical distribution, the Concrete Mixture model, introduced in 2016. Although the idea appears attractive, a straightforward implementation using the No-U-Turn Sampler Hamiltonian Monte Carlo of the Stan software offers poor performance. I am not overly surprised as, in my own limited experience, continuous relaxations of discrete problems create rapidly varying gradients which thwart the Stan implementation of HMC. Tempering or careful tuning and initialisation might improve performance, but this might be out of the scope of the present contribution.

Here are some comments on the manuscript:
- The comparison to the other algorithms may be too severe for the Concrete Mixture approximation:
          - The marginalised model can be a good reference point, but it samples from a simpler posterior and benefits from Rao-Blackwellisation, so better performances could be expected
          - The Gibbs model works well in small dimension, I would expect the benefit of HMC with respect to Gibbs to kick in when exploring much higher dimensional spaces. The toy benchmark benefits Gibbs versus HMC
          - A proper comparison might have been Nishimura et al. (2020), which does not marginalise but still uses HMC and might scale better than Gibbs in high dimension.
- The concrete mixture could be explained a bit better, maybe with a graph showing the effect of the temperature parameter. There is probably a constraint that pi lies in the simplex, which is missing from eq. (1), and probably another constraint on the sum of the zi variables.
- In equation (3), you could write that this is the marginalised version of the posterior
- In equation (4), the definition of (c) the allocation variables is missing, you should write that it’s a vector of integers denoting cluster membership
- The definition of the estimator for c, termed a majority vote, could probably called a marginal MAP estimator, which might make it sound more natural
- In table A, Acc_c is not defined. What is this actually, an acceptance rate when updating the c vector? If so, isn’t it too high?
- In Table 1, you would want to see the effective sample size per second rather than runtime. Otherwise, there is no relevant information available.
- Equation (8) does not seem to make much sense, why do you not use the same estimator c as before, what was called the majority vote?
- Finally, one of the most important comments: using Mean Squared Error as a measure of performance does not make sense. It assumes that a good algorithm or a good approximation would recover the hidden parameters exactly. This is not Bayesian, unless there were a very large amount of data, there is no reason for the posterior to be exactly concentrated on the simulation parameters. A proper measurement of performance would be: are you able to recover the correct posterior using the Concrete Mixture approximation? The correct posterior can be obtained, for instance, by running the Gibbs sampler for many iterations, then there are multiple ways to measure the distance between the approximate posterior obtained via the Concrete Mixture and the correct posterior. This way, you could assess the deterioration of the approximation when increasing the temperature and look for a compromise with the improvement of the ESS/s

---

### Official Review · Reviewer_WF9Q · 2024-05-02
**Simple but useful findings**

**Rating:** 6
**Confidence:** 4

**Review:**

The paper studies how to use the Concrete distribution to replace the categorical distribution in models so that (standard) HMC can be used to sample from the posterior directly.
The paper focuses on the Concrete mixture model as the correspondence of mixture model where the categorical distribution is replaced by the Concrete distribution and conducts simulated experiments on data simulated from a mixture of Gaussian.

## quality
The method and the findings in the paper are sound.

## clarity
The paper is well-written and easy to follow.
Key citations are provided.

## originality & significance
The paper mainly applies an existing technique to a known problem
However, even the idea behind this paper has been there in the community for a while but I didn't read work on studying and discussing the usefulness/limitations of this idea.
The paper addressed one question I've been wondering for a while which is how much bias the continuous relaxation would lead to (with different temperature parameters).
I also really like this paper as it points out the idea works, but with limitations on the efficiency (both statistically as the posterior gets more difficulty to sample with lower $\tau$, and computationally as there more parameters to work with).
I think this is a useful reading for people to study this problem and can potentially motivates future work, e.g. would Riemannian HMC help with the more difficult geometry?
In short I think it's a useful reading as a first-step to find the right combinations of parameters/techniques to make this idea really work (in practice, for a wide-range of problems).

---

### Meta-Review · Area_Chair_RPRf · 2024-05-26

**Recommendation:** Accept (Poster)
**Confidence:** 4

**Metareview:**

All reviewers agree that this is a good paper and should be presented at the symposium. Having read the reviews and the paper, I agree. The reviewers have shared detailed feedback, and I strongly recommend that the authors incorporate the reviewers' feedback.

---

### Decision · Program_Chairs · 2024-05-27

Accept